# Ufmylation of UFBP1 Is Dispensable for Endoplasmic Reticulum Stress Response, Embryonic Development, and Cardiac and Intestinal Homeostasis

**DOI:** 10.3390/cells12151923

**Published:** 2023-07-25

**Authors:** Varsha Tandra, Travis Anderson, Juan D. Ayala, Neal L. Weintraub, Nagendra Singh, Honglin Li, Jie Li

**Affiliations:** 1Vascular Biology Center, Medical College of Georgia, Augusta University, Augusta, GA 30912, USA; 2Division of Cardiology, Department of Medicine, Medical College of Georgia, Augusta University, Augusta, GA 30912, USA; 3Department of Biochemistry and Molecular Biology, Medical College of Georgia, Augusta University, Augusta, GA 30912, USA

**Keywords:** ufmylation, UFM1, UFBP1, ER stress response, heart failure, intestine

## Abstract

Protein modification by ubiquitin fold modifier 1 (UFM1), termed ufmylation, regulates various physiological and pathological processes. Among emerging UFM1 targets, UFM1 binding protein 1 (UFBP1) is the first identified ufmylation substrate. Recent clinical and animal studies have demonstrated the pivotal roles of UFBP1 in development, hematopoiesis, intestinal homeostasis, chondrogenesis, and neuronal development, which has been linked to its function in maintaining endoplasmic reticulum (ER) homeostasis. However, the importance of UFBP1 ufmylation in these cellular and physiological processes has yet to be determined. It has been proposed that ufmylation of lysine 268 (267 in *humans*) in UFBP1 plays a critical role in mediating the effects of the ufmylation pathway. In this study, we for the first time probe the pathophysiological significance of UFBP1 ufmylation in vivo by creating and characterizing a *mouse* UFBP1 knockin (KI) model in which the lysine 268 of UFBP1, the amino acid accepting UFM1, was mutated to arginine. Our results showed that the K268R mutation reduced the total ufmylated proteins without altering the expression levels of individual ufmylation enzymes in mouse embryonic fibroblasts. The K268R mutation did not alter ER stress–stimuli–induced ER stress signaling or cell death in mouse embryonic fibroblasts. The homozygous KI mice were viable and morphologically indistinguishable from their littermate wild–type controls up to one year of age. Serial echocardiography revealed no cardiac functional impairment of the homozygous KI mice. Furthermore, the homozygous KI mice exhibited the same susceptibility to dextran sulfate sodium (DSS) –induced colitis as wild-type mice. Taken together, these results suggest that UFBP1 K268 is dispensable for ER stress response, embryonic development, cardiac homeostasis under physiological conditions, and intestinal homeostasis under pathological conditions. Our studies call for future investigations to understand the biological function of UFBP1 ufmylation and offer a new mouse model to determine the roles of UFBP1 ufmylation in different tissues under stress conditions.

## 1. Introduction

A basic understanding of post_translational modifications (PTMs), their distinct mechanisms, and their regulation of numerous cellular, physiological, and pathological processes is quickly evolving. Ufmylation is an evolutionarily highly conserved ubiquitin-like protein modification by which the ubiquitin–like protein UFM1 (ubiquitin fold modifier 1) covalently modifies protein substrates [1,2]. Similar to ubiquitination, ufmylation is catalyzed by an E1–E2–E3 enzymatic cascade comprising E1 activating enzyme UBA5, E2 conjugating enzyme UFC1, and E3 ligase enzyme UFL1 [3]. This event results in the fusion of the C–terminal glycine of UFM1 to lysine residues on substrate proteins [1]. Ufmylation is reversible. Deconjugation of the UFM1 molecule is carried out by the same proteases UFSP1 and UFSP2 [4]. Recent clinical and animal studies have uncovered an important role of ufmylation in health and disease. Polymorphisms and mutations of different ufmylation components have been linked to encephalopathy [5], Beukes familial hip dysplasia [6], and cancer [7]. Moreover, genetic studies using animals with deletion of different ufmylation enzymes have demonstrated the pivotal roles of ufmylation in neuronal development [8], hematopoiesis [9], tumor progression [10], heart failure [11], diabetes [12], and viral infection [13]. At cellular levels, ufmylation participates in ER homeostasis [14], cell differentiation, and cell cycle control [15].

Our understanding of the biological function of ufmylation is rapidly expanded by the discovery of novel UFM1 targets such as UFBP1 (UFM1 binding protein 1, also known as DDRGK1), ASC1, and RPL26 [16,17,18]. Among them, UFBP1 was the first identified UFM1 substrate [19,20]. Ufmylation of UFBP1 occurs mainly on its lysine 267 in humans (lysine 268 in mice), although the possibility of other lysine residues required for ufmylation is not negligible [19]. UFBP1 localizes to the ER membrane via an N–terminal signal peptide [21]. Besides its role as a substrate, UFBP1 has been recognized as an interacting partner and an adaptor of UFL1, which is proposed to be important for its localization [9,17] or its UFL1 ufmylation activity [22]. Mutations of UFBP1 are associated with skeletal dysplasia [23]. Deletion of the *UFBP1* gene impairs hematopoiesis [9], plasma cell development [24], intestinal homeostasis [25], chondrogenesis [23], and neuronal cell survival [26], highlighting its pivotal roles in tissue homeostasis. 

Despite the importance of ufmylation in health and disease and the expanding list of UFM1 targets, the pathophysiological significance of the ufmylation of any specific UFM1 target has not been elucidated in vivo. Further, as the first identified UFM1 target, it remains controversial whether UFBP1 acts as a UFM1 substrate or a component of ufmylation machinery to exert its biological function. To test the former hypothesis, we created and characterized a *mouse* UFBP1 knockin (KI) model in which UFBP1 ufmylation was abolished by mutating the main ufmylation site lysine 268 to arginine (K268R). Our results showed that K268R mutation inhibits protein ufmylation but has no impact on ER stress responses, embryonic development, cardiac function, and experimentally induced colitis. These findings for the first time demonstrate a dispensable role of UFBP1 ufmylation in development and cardiac and intestinal homeostasis and therefore call for future investigations to define the biological function of UFBP1 ufmylation in vivo under both physiological and pathological conditions.

## 2. Methods

Animal studies UFBP1^K268R^ KI mice: UFBP1^K268R^ KI transgenic mice with K→R mutation at position 268 were generated by the CRISPR–Cas9 system. The specific gRNA–TTATGTAGATAAACTTGCCC CGG was designed to introduce a mutation at K268 to R. Genotyping of UFBP1^K268R^ knockin and wild-type alleles was performed by genomic DNA sequencing with sequencing primer AGAGCCAAGCCTACAAACAG. 

All animal experiments were approved by the Augusta University Institutional Animal Care and Use Committee. During all experiments involving transgenic mice, experimental mice were randomly assigned into all groups, including sex (male and female) and age (8–12 weeks for breeding), unless specified. All mice were at healthy status before the experiments. The housing conditions were well maintained at room temperature in the animal facility of Augusta University. The influence of sex was not determined for the sacrificed mice.

Isolation of MEFs Mouse embryonic fibroblasts were harvested from UFBP1 WT and homozygous KI as previously described [27]. Briefly, mouse embryos were collected from pregnant females at 14.5 days of gestation. The embryos were dissected, and the cells were isolated and cultured in DMEM with 10% FBS and 1% pen/strep.

Cell Treatments MEF WT and UFBP1 homozygous KI cells were treated with known ER stress inducer tunicamycin (1 µg/mL) (Sigma–Aldrich, Saint Louis, MO, USA) for 6 or 24 h, followed by RNA and protein extraction, respectively.

Western blotting Protein samples for Western blot were processed as described previously [28]. Briefly, the protein lysates were extracted from mouse tissue and cultured cells. Protein concentrations were determined by BCA followed by SDS–PAGE, immunoblotting, and densitometry analysis. Frozen tissues or cells were homogenized in lysis buffer (50 mm Tris–HCl pH 6.8 containing 1% SDS, 10% glycerol, and complete protease inhibitor mixture), sonicated, and spun down at 12,000 rpm. The supernatant was collected, and the protein concentration was determined. The protein lysate was mixed with half the volume of 3x SDS loading buffer with 15% β–mercaptoethanol. The mixed sample was next boiled for 10 min and subjected to SDS–PAGE gel isolation, then transferred to a PVDF membrane. The antibodies used in this study are listed below.
**Protein****Company****Cat#**UFM1Abcam10862UBA5Proteintech12093-1-APUFC1Proteintech15783-1-APUFL1Bethyl LaboratoriesA303-456AUFBP1Proteintech21445-1-APUFSP2Proteintech16999-1-APRPL26Cell signaling5400SCDK5RAP3/C53Proteintech11007-1-APPERKCell signaling3192p-PERKCell signaling3179SIRE1aCell signaling3294XBP1sCell signaling12782ATF4Cell signaling11815SCHOPCell signaling2895S

Quantitative real time PCR RNA isolated from cultured cells were reverse–transcribed into single–stranded cDNA as previously described [28]. Gene expression levels were measured in at least triplicate per sample by real–time quantitative PCR (QuantStudio 3 Real–Time PCR system, Thermo Fisher Scientific, Waltham, MA, USA) using the SYBR–Green assay with gene–specific primers at a final concentration of 200 nM. The primers used for qPCR are listed below. Relative gene expression was calculated using the 2−∆∆ct method against a mouse housekeeping gene, *Rplp0*. Each experiment was repeated at least three times independently. Primers used in this study are as follows:
**Mouse****Forward Primer****Reverse Primer***Ufm1*GGAATAGGAATAAATCCTGCACACAGTTCTGAGCCGTGCTTC*Uba5*GTACCACAGCGACAGAAAGACAGTCAGTCCATCCTCCATTT*Ufc1*AGTAGCTGGAGGGACAAATAAGGCCTATTTCAAGGCCCAAATC*Ufl1*TCATCAACCACGGGCATAAAGTTTCCTCTCCACTCTCCTTTC*Ufbp1*AGAACAGAAGGAGGAGGAAGAGCTCAGTCATGGTTTCGCTAACA*Ufsp2*CTTGCACGAAGTAGGTGTTCTATCGAAGCCGCCTCTTATTC*Rpl26*GGAATAGGAATAAATCCTGCACACAGTTCTGAGCCGTGCTTC*Cdk5rap3/C53*CGCTCTGACTCTTCTGGAATACCCTCCTCGCTCATCTCTACT*Rplp0*CCTCCTTCTTCCAGGCTTTGCCACCTTGTCTCCAGTCTTTATC*Bip*GCTTCGTGTCTCCTCCTGAC
GGAATAGGTGGTCCCCAAGT*Pdi*GCCCAAGAGTGTATCTGACTATGGGTTATCAGTGTGGTCGCTATC*Chop*GTCCTGTCCTCAGATGAATTGGGCAGGGTCAAGAGTAGTGAAGGTT*Atf4*GGAATAGGAATAAATCCTGCACACAGTTCTGAGCCGTGCTTC*PerK*CCC AGG CAT TGT GAG GTA TTCCA GTC TGT GCT TTC GTC TT*Xbp1s*GAGTCCGCAGCAGGTGGTGTCAGAGTCCATGGGA

Cell death assays Cell death analysis was performed by the measurement of lactose dehydrogenase (LDH) released from damaged cells to the cell culture media using a cytotoxicity detection kit (Roche, Mannheim, Germany). Using the LIVE/DEAD^TM^ Viability/Cytotoxicity Kit (Thermo Fisher Scientific, West Columbia, SC, USA), a two–color fluorescence cell viability assay was measured in the MEFs. After staining, the cells were gently washed with 1x PBS and imaged by fluorescence microscopy. The percentage of live to dead cells was quantified by ImageJ software, 1.54d, National Institutes of Health, USA. A total of seven views/sample, ~1000 cells per group, were quantified. 

Echocardiography All the mice were anesthetized by inhalation of isoflurane (2.0% for induction and 1.2% for maintenance) via a nose cone. The adequacy of anesthesia was monitored by toe pinch. Cardiac B–mode images and M–mode loops were recorded using a VEVO 2100 echocardiography system with a 30 MHz transducer (Visual Sonics). The LV morphometric and functional parameters were analyzed offline using VEVO 2100 software. Mice genotyping information was blind to the echocardiography performer.

DSS treatment experimental procedures The wild type(+/+), UFBP1^K268R^ KI/+, and UFBP1^K268R^ KI/KI mice were fed with 5% DSS (Thermo Fisher Scientific, West Columbia, SC, USA) in autoclaved drinking water for five days, and the control mice were fed with normal autoclaved drinking water. The mice were monitored daily for body weight, clinical score for diarrhea, and anal appearance. After five days, the mice were fed with autoclaved drinking water for two additional days. The following parameters were considered for scoring: diarrhea: 0 = normal stool with negative hemoccult, 1 = soft stools with positive hemoccult, 2 = very soft stools with traces of blood, 3 = watery stools with visible rectal bleeding, 4 = watery stools with excessive bleeding; anal appearance: 0 = normal, 1 = slightly reddish and swelling, 2 = bleeding. The mice were sacrificed after the seventh day, and gravimetric data were collected as per IACUC guidelines.

Quantification and statistical analysis All statistical analysis was performed with GraphPad Prism 9.4.1, Dotmatics (Boston MA, USA) software. Results are shown as mean ± SEM. An unpaired *t*-test was used to compare two specific groups. For multiple comparisons, one-way ANOVA followed by post hoc Tukey’s multiple comparison test or, when appropriate, two–way ANOVA, was performed. A *p*-value < 0.05 was considered statistically significant. In the figures, different *p* values were labeled as follows: * 0.05 > *p* ≥ 0.01; ** 0.01 > *p* ≥ 0.001, *** 0.001 > *p* ≥ 0.0001; **** *p* < 0.0001. Specific statistical analysis methods and the n number of individual experiments can be found in the related figure legend. All error bars in this article represent ± SEM unless stated otherwise in figure legends.

## 3. Results

### 3.1. Generation of UFBP1 K268R KI Mice

UFBP1 was the very first substrate identified in the ufmylation pathway, and the amino acid at K267 was reported to be the possible site for ufmylation [19]. 

To address the function of the ufmylation of UFBP1 in vivo, we generated UFBP1 K268R KI mice, via CRISPR–Cas9 technology (Figure 1A), in which the lysine 268 (equivalent to lysine 267 in *humans*) of UFBP1 was mutated to arginine. Specifically, we designed a specific guided RNA that introduces a missense mutation (A→G, mutating K268 to R) to the *Ufbp1* locus and a synonymous mutation (C→T, encoding D265). The latter can avoid inaccurate re–cleavage of the edited DNA by Cas9. The mutation was confirmed by DNA sequencing using genomic DNA (Figure 1B). Western blot further showed that the levels of UFBP1 proteins were comparable in all tested tissues between wild–type (WT) and homozygous KI mice (Figure 1C), suggesting that the mutation has no impact on UFBP1 expression.

### 3.2. The Impact of K268R Mutation on Ufmylation 

To determine the effect of the mutation on ufmylation, we isolated primary mouse embryonic cells (MEF cells) from homozygous KI mice and littermate WT mice. Western blot of the protein extracts from MEFs showed unchanged levels of free UFM1 proteins between WT and homozygous KI MEFs (Figure 2A,B). Interestingly, we observed a significant reduction of multiple ufmylated proteins (band 2 and 3 in Figure 2A) in UFBP1 KI MEFs. However, the mutation did not seem to affect the expression of ufmylation enzymes including UBA5, UFC1, UFL1, and UFSP2, in addition to the ufmylation substrate RPL26 and a UFBP1–interacting protein C53. Together, these data suggest that UFBP1 ufmylation is required for the ufmylation of a subset of unknown proteins.

### 3.3. The Effect of K268 Mutation on ER Stress Response

Previous studies demonstrated a key role of UFBP1 in the regulation of ER stress response, in part through its remaining controversial impact on IRE1α–XBP1 signaling response [29].To investigate the effect of K268R mutation on ER homeostasis, we probed the responses of WT and KI MEFs to an ER stress inducer, tunicamycin (TM). Our qPCR analyses showed the equivalent upregulation of ER chaperons (*Bip*, *Pdi*, and *Chop*) and ER stress transducers/effectors (*Atf4*, *Perk* and *Xbp1s*) in WT and KI MEFs (Figure 3A). Consistently, Western blot revealed that K268R mutation did not alter the expression of PERK and IRE1α and did not alter TM–induced phosphorylation of PERK or upregulation of ATF4, XBP1s, and CHOP (Figure 3B). We further examined ER–stress–induced cell death. As expected, ER stress inducers elevated lactate dehydrogenase (LDH) activity in the culture medium and increased cell death (Figure 3C–E). However, none of these events was affected by K268R mutation (Figure 3C–E). Taken together, our results suggest that the K268 of UFBP1 is not required for activation of UPR signaling and ER–stress–induced cell death.

### 3.4. Ufmylation of UFBP1 Is Not Essential for Development

Germline and inducible deletion of *Ufbp1* causes lethality at the embryonic and adult stage, respectively [9]. The heterozygous KI mice were viable and fertile and were then crossed to generate homozygous KI mice. The homozygous KI mice were born at the expected Mendelian ratio. These mice were morphologically indistinguishable from littermate WT mice at birth (Figure 4A) and adulthood (Figure 4B). We did not see any growth abnormality in homozygous KI mice when measuring the body weight and organ weight by seven months of age (Figure 4C,D). These data suggest that ablation of UFBP1 ufmylation has no impact on development.

### 3.5. Role of Ufmylation of UFBP1 on Cardiac Function

Cardiac–specific ablation of UFL1 led to heart failure with a concomitant reduction of UFBP1 [11], implying a possible role of UFBP1 in the heart. By seven months of age, the homozygous KI hearts were morphologically normal and had a comparable heart weight to body weight ratio to littermate WT mice (Figure 5A,B). Echocardiography showed a comparable left ventricular ejection fraction, left ventricular fractional shortening, systolic posterior wall thickness, and left ventricular diastolic internal chamber diameter among WT, heterozygous, and homozygous KI mice (Figure 5C), indicative of intact cardiac chamber development and normal cardiac contractility.

### 3.6. Ufmylation of UFBP1 and Intestinal Homeostasis 

Intestinal epithelial–specific UFL1 and UFBP1 knockout mice were more susceptible to experimentally induced colitis [25]. To determine the possible role of UFBP1 ufmylation in gut homeostasis, we treated seven–month–old WT and heterozygous and homozygous KI mice with dextran sodium sulphate (DSS), a chemical colitogen with anticoagulant properties that is widely used to establish mouse models of colitis (Figure 6A). As expected, DSS induced a gradual loss of body weight (Figure 6B), bleeding of the anus (Figure 6C), diarrhea as indicated by soft stool (Figure 6D), and reduction of colon length (Figure 6E,F) in all three groups of mice. However, neither heterozygous nor homozygous KI mice exhibited a deteriorated clinical score for anal appearance, diarrhea, or colon shrinkage. These findings suggest that UFBP1 ufmylation is not essential for preventing inflammatory colitis.

## 4. Discussion

Using MCF7 and HepG2 cells, it was shown that ufmylation of UFBP1 at K267 plays an important role in its interaction with IRE1α and stability [29]. UFBP1 has been highlighted as an important substrate and a co–factor of the ufmylation pathway regulating ER homeostasis, cell cycle, and cell differentiation [20,24]. Therefore, it is of great interest to explore the possible role of the ufmylation of UFBP1 in its functions. Using K268R KI MEFs and mice, this study for the first time reported the functional significance of the lysine 268 of UFBP1 in vivo. Our in vitro and in vivo findings demonstrate that although UFBP1 controls ER stress response [20,24], embryonic and postnatal development [9,30], and gut homeostasis [25], ufmylation of UFBP1 at lysine 268 is not required for its physiological and pathological function under tested conditions. 

An interesting observation of this study is the reduction of ufmylated proteins by UFBP1 K268R mutation (Figure 2). Notably, this mutation does not alter the expression of key ufmylation enzymes. How exactly the ufmylation enzymes catalyze the modification of protein targets by UFM1 is not understood. A possible explanation for this observation is that UFBP1 may act as a component of ufmylation machinery [22] and that ufmylation of UFBP1 may induce the assembly of a ufmylation enzyme complex to promote ufmylation. Despite extensive efforts, we were not able to detect the interaction of endogenous UFBP1 with UFL1, C53, and RPL26, three reported UFBP1 interacting partners [18,19,21,22], by immunoprecipitation using MEFs in our studies, possibly due to weak, transient, and dynamic interactions. Therefore, it remains to be tested whether ufmylation of UFBP1 regulates its binding affinity to other proteins. 

While the knockout of UFBP1 in mice led to significant phenotypic changes, including development defects and exacerbated colitis [25], the absence of the ufmylation of UFBP1 did not appear to have similar effects. This result suggests that the biological function of UFBP1 is not necessarily reliant on its ufmylation. In line with our observations, structure and function analysis suggest that the lysine 267 of UFBP1 is also dispensable for the development of plasmablasts but is required for immunoglobulin production and stimulation of ER expansion in IRE1α–deficient plasmablasts [24]. These lines of evidence favor the concept that UFBP1 acts as part of the ufmylation machinery instead of the downstream effectors of ufmylation. It should be pointed out that although K268 was originally identified as the main ufmylation site on UFBP1, other lysines (116, 121, 124, 128, 193, 224, and 227) on UFBP1 can also accept UFM1 [24]. As we learned from the case of ubiquitin and ubiquitin–like proteins, the modification sites can shift from one to another, especially when one site is blocked [2]. Therefore, we cannot rule out the possibility that K268R may trigger compensatory ufmylation of UFBP1 on other lysine residues, leading to intact UFBP1 functions. 

In summary, our seemingly “negative” findings prove the dispensable role of UFBP1 ufmylation in the maintenance of ER homeostasis, development, cardiac function, and gut homeostasis. More research efforts are needed to uncover the significance of the ufmylation of UFBP1 in the UFM1 cascade and the role of the ufmylated form of UFBP1 in regulating various cellular and pathophysiological events. 

## Figures and Tables

**Figure 1 cells-12-01923-f001:**
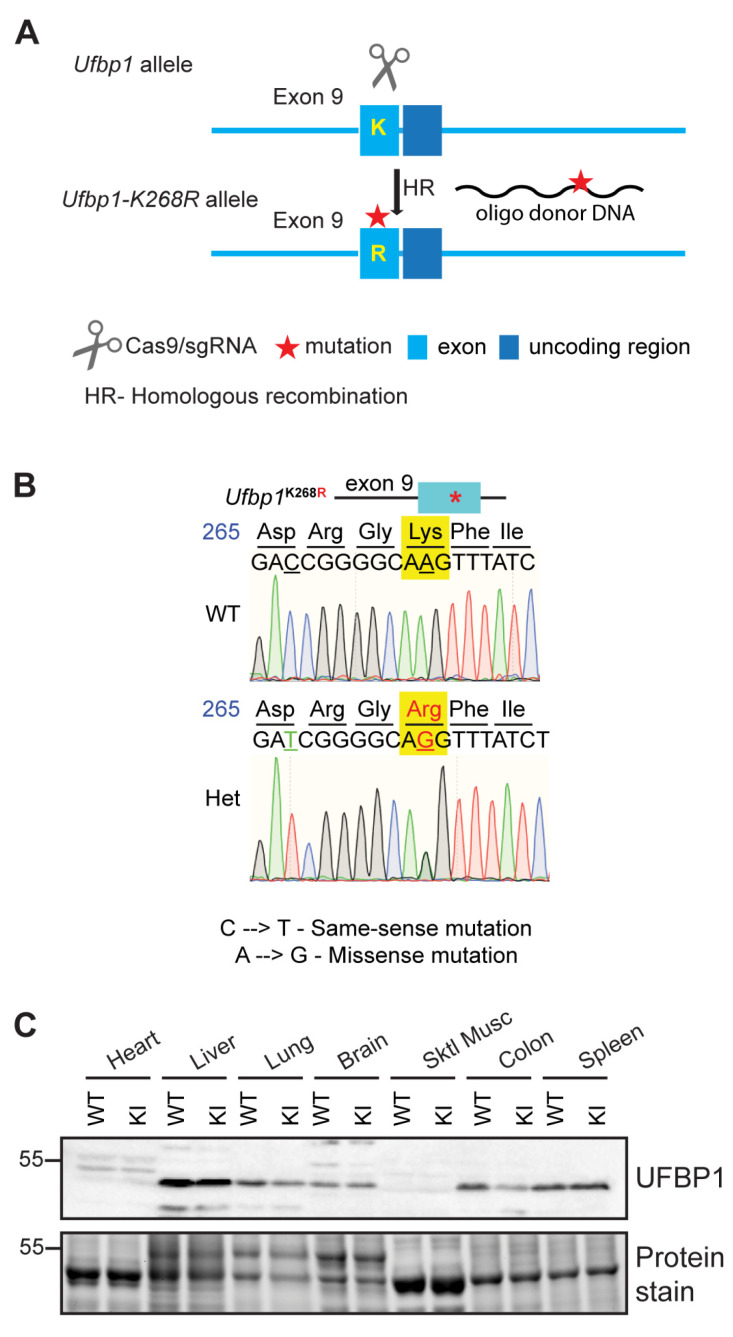
Creation of UFBP1 K268R knockin (KI) mice. (**A**) Schematics of the generation of UFBP1 K268R mice via CRISPR–Cas9 gene editing strategy. (**B**) Validation of K268R mutation in the KI allele by genomic DNA sequencing. (**C**) Western blot analysis of UFBP1 protein levels in different organs from seven-month-old WT and homozygous (KI) mice.

**Figure 2 cells-12-01923-f002:**
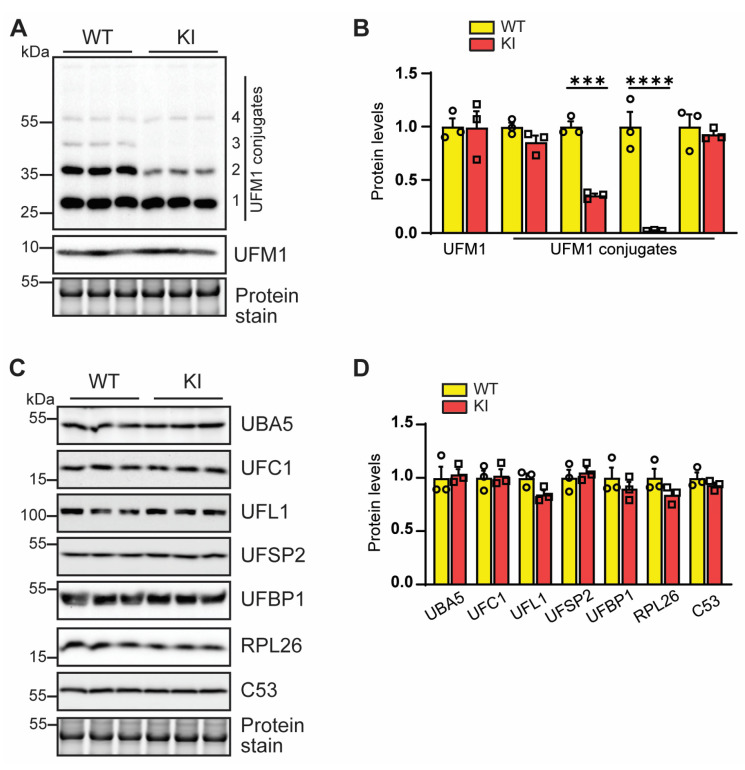
The impact of UFBP1 K268R KI on ufmylation. (**A**,**B**) Western blot analysis (**A**) and quantification (**B**) of UFM1 and UFM1 conjugates in total protein lysates from WT and homozygous KI MEFs, *n* = 3. (**C**,**D**) Western blot analysis (**C**) and quantification (**D**) of ufmylation pathway components in MEFs, *n* = 3. *** *p* < 0.001; **** *p* < 0.0001. Error bars indicate SEM. Two-way ANOVA, followed by multiple comparisons, was performed.

**Figure 3 cells-12-01923-f003:**
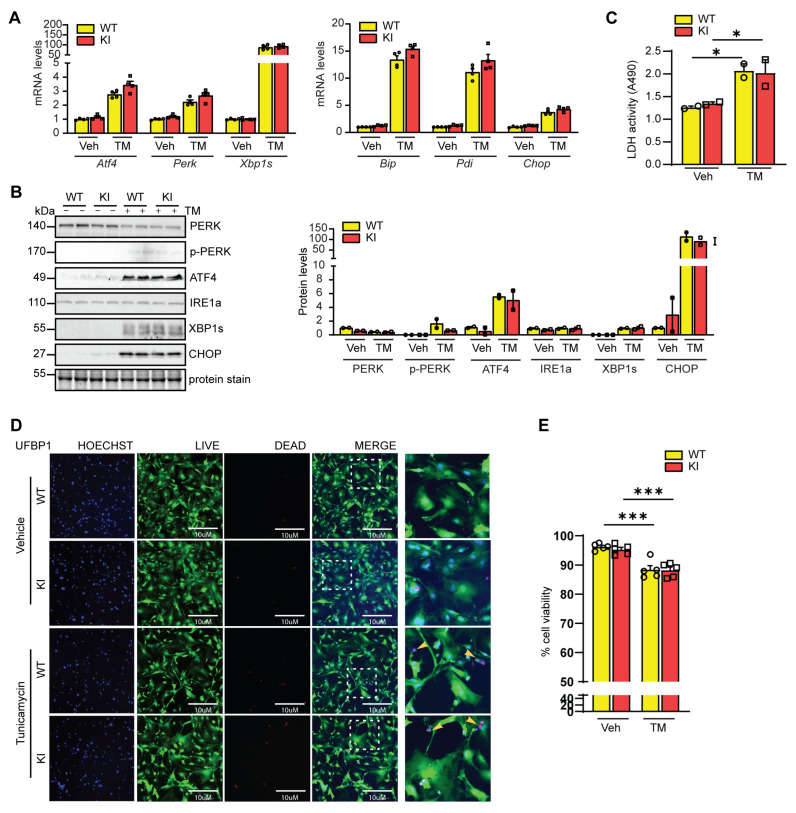
K268R mutation has no impact on ER stress response. WT and homozygous KI MEFs were treated with tunicamycin (1µg/mL) for 6 h (**A**,**B**) or 24 h (**C**,**D**) before the harvest for subsequent analyses. (**A**) The qPCR analyses of indicated ER stress effectors, *n* = 4. (**B**) Western blot (**B**) and quantification of indicated proteins, *n* = 2. (**C**) Lactate dehydrogenase (LDH) activity assay, *n* = 2. Live (green)–Dead (red) viability/cytotoxicity assay (**D**,**E**). (**D**) Representative images of live and dead cell staining. Nuclei were counterstained with Hoechst 33342 (blue). White dotted line indicates the zoom image of selected area. The yellow arrowheads indicate dead cells. Scale bar, 10 µm and quantification of percentage viable cell (**E**). 7 areas, ~1000 cells per group were counted. Unpaired *t*–test. * *p* < 0.05 considered significant; *** *p* < 0.001. Data mean ± SEM.

**Figure 4 cells-12-01923-f004:**
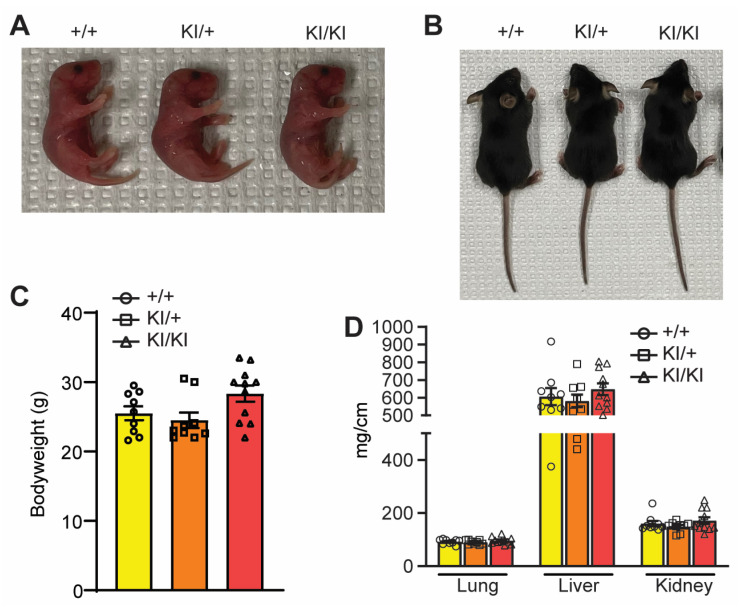
K268R mutation does not influence embryonic development. (**A**) Gross morphology of indicated mouse neonates at postnatal Day 3. (**B**) Gross morphology of indicated mice at 2 weeks of age. (**C**) Bodyweight at seven months old, *n* = +/+ = 9, KI/+ = 9, and KI/KI = 11. (**D**) Major organs—lung, liver, and kidney–weight normalized to tibia length. One–way ANOVA followed by post hoc Tukey’s multiple comparison test was performed. *p* < 0.05 considered significant. Data mean ± SEM.

**Figure 5 cells-12-01923-f005:**
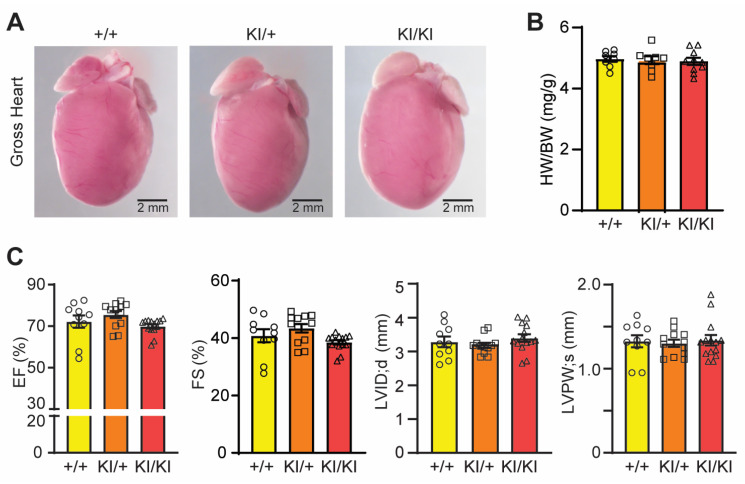
K268R mutation does not cause cardiac dysfunction during aging. (**A**) Gross morphology of hearts from seven–month–old hearts. (**B**) Heart weight to body weight ratio of hearts at seven months of age, *n* = +/+ = 8, KI/+ = 8, and KI/KI = 10. (**C**) Quantification of echocardiographic data at seven months of age. EF, ejection fraction; FS, fractional shortening; LVPWs, systolic left ventricular posterior wall thickness; LVIDd, diastolic left ventricular internal diameter, *n* = +/+ = 10, KI/+ = 12, and KI/KI = 14. One–way ANOVA followed by post hoc Tukey’s multiple comparison test was performed. *p* < 0.05 considered significant. Data mean ± SEM.

**Figure 6 cells-12-01923-f006:**
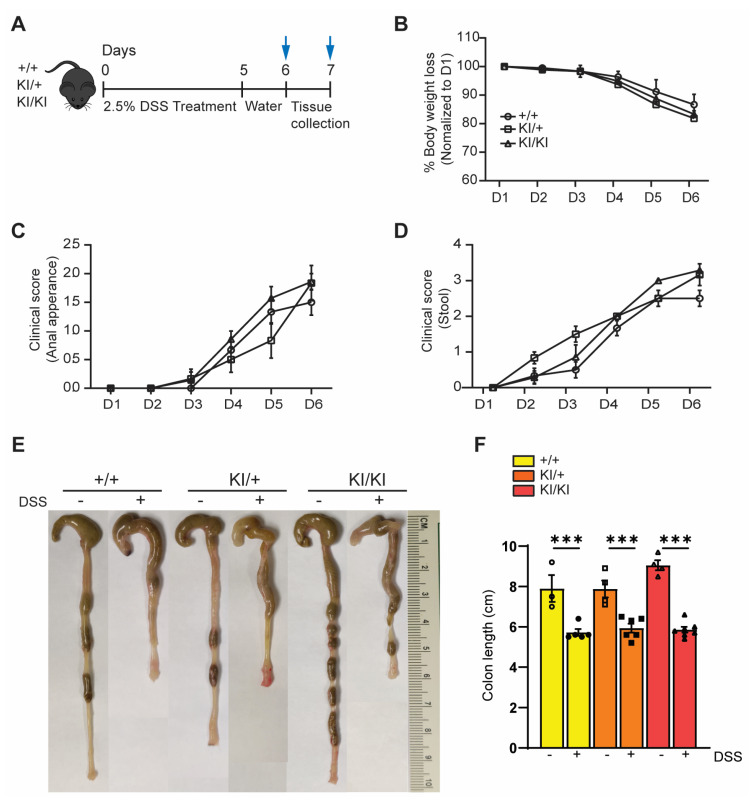
K268R mutation does not exacerbate DSS–induced colitis. (**A**) Scheme of experimental design. Seven–month–old mice were treated with 5% DSS in drinking water for 5 consecutive days and traced for an additional 2 days. (**B**) Changes in body weight over time. The percentage of body weight loss was calculated relative to the body weight of the individual mouse group at Day 0. (**C**,**D**) Clinical scores of anal appearances (**C**) and diarrhea (**D**) of DSS–treated mice, *n* = +/+ = 6, KI/+ = 6, and KI/KI = 7. (**E**) Representative gross morphology of the colon with and without DSS in each group. (**F**) Quantification of gastrointestinal length. One–way ANOVA followed by post hoc Tukey’s multiple comparison test was performed. Data mean ± SEM. *** *p* < 0.001.

## Data Availability

Data is unavailable.

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
