# Peer review of "Ufmylation of UFBP1 Is Dispensable for Endoplasmic Reticulum Stress Response, Embryonic Development, and Cardiac and Intestinal Homeostasis"

_cells, 2023, doi:10.3390/cells12151923_

Round 1
Reviewer 1 Report
Through different experiments using transgenic mice with K -> R mutation at position 268 of UFBP1, the authors show that ufmylation is not essential in a number of physiological processes, which include ER homeostasis, development, cardiac function, and gut 233 homeostasis.
A few concerns are suggested to be conveniently addressed:
1) Lines 90 - 95, and 103 - 104: these paragraphs should be removed as they do not conform a description of results. They should be rather included in the discussion section
2) Lines 118 - 119: As in point 1, but additionally it should be better explained what the requirements are for "overall ufmylation" and where the step of "UFBP1 ufmylation" is placed.
3) Lines 129 - 130: "we probe the 129 responses of WT and KI MEFs to two ER stress inducer, tunicamycin (TM)". Please rewrite.
4) Lines 230 - 231: why was not experimentally investigated the possible ufmylation of other lysines? Would not other lysine ufmylation be in contradiction with the specifity of this particular protein modification process, in contrast to what happens with ubiquitination? Why would other lysines be ufmylated?
Author Response
Reviewer 1
Through different experiments using transgenic mice with K -> R mutation at position 268 of UFBP1, the authors show that ufmylation is not essential in a number of physiological processes, which include ER homeostasis, development, cardiac function, and gut homeostasis.
A few concerns are suggested to be conveniently addressed:
1) Lines 90 - 95, and 103 - 104: these paragraphs should be removed as they do not conform a description of results. They should be rather included in the discussion section.
We appreciate the suggestions. We have now moved Lines 90-95 to the beginning of the discussion as follows:
“Using MCF7 and HepG2 cells, it was shown that ufmylation of UFBP1 at K267 plays an important role in its interaction with IRE1a and stability27. UFBP1 has been highlighted as an important substrate and a co-factor of the ufmylation pathway regulating ER homeostasis, cell cycle and cell differentiation20, 24. Therefore, it is of great interest to explore the possible role of ufmylation of UFBP1 in its functions. Using K268R KI MEFs and mice, this study for the first time reported the functional significance of lysine 268 of UFBP1 in vivo. Our in vitro and in vivo findings demonstrate that although UFBP1 controls ER stress response24, 20, embryonic and postnatal development 9, 28, and gut homeostasis25, ufmylation of UFBP1 at lysine 268 is not required for its physiological and pathological function under tested conditions.
”
With regard to Line 103-104 'suggesting that the mutation has no impact on UFBP1 expression,' we believe it is necessary to provide a brief interpretation of the presented data. Therefore, we have decided to retain this sentence in its current placement.
2) Lines 118 - 119: As in point 1, but additionally it should be better explained what the requirements are for "overall ufmylation" and where the step of "UFBP1 ufmylation" is placed.
We appreciate the comments. We agree that it is necessary to include a discussion on how UFBP1 ufmylation could potentially affect protein ufmylation. The discussion on this data was included in Line 203-213 as follows:
“An interesting observation of this study is the reduction of ufmylated proteins by UFBP1 K268R mutation (Figure 2). Notably, this mutation does not alter the expression of key ufmylation enzymes. How exactly the ufmylation enzymes catalyze the modification of protein targets by UFM1 is not understood. A possible explanation for this observation is that UFBP1 may act as a component of ufmylation machinery22 and that ufmylation of UFBP1 may induce the assembly of ufmylation enzyme complex to promote ufmylation. Despite extensive efforts, we were not able to detect the interaction of endogenous UFBP1 with UFL1, C53 and RPL26, three reported UFBP1 interacting partners19, 21, 22, 18 , by immunoprecipitation using MEFs at our hands, possibly due to the weak, transient and dynamic interaction. Therefore, it remains to be tested whether ufmylation of UFBP1 regulates its binding affinity to other proteins”
Meanwhile, to improve the accuracy, we have revised Line 118-119 to “Together, these data suggest that UFBP1 ufmylation is required for the ufmylation of a subset of unknown proteins.”
3) Lines 129 - 130: "we probe the 129 responses of WT and KI MEFs to two ER stress inducer, tunicamycin (TM)". Please rewrite.
We appreciate for pointing out the mistake. This sentence is now corrected to “we probe the responses of WT and KI MEFs to two ER stress inducer, tunicamycin (TM)”
4) Lines 230 - 231: why was not experimentally investigated the possible ufmylation of other lysines? Would not other lysine ufmylation be in contradiction with the specifity of this particular protein modification process, in contrast to what happens with ubiquitination? Why would other lysines be ufmylated?
We appreciate the comments. These data were discussed in the manuscript (Lines 214-227) as follows:
“It should be pointed out that although K268 was originally identified as the main ufmylation site on UFBP1, other lysines (116, 121, 124, 128, 193, 224, and 227) on UFBP1 can also accept UFM124. As we learned from the case of ubiquitin and ubiquitin-like proteins, the modification sites can shift from one to another, especially when one site is blocked2. Therefore, we cannot rule out the possibility that K268R may trigger compensatory ufmylation of UFBP1 on other lysine residues, leading to intact UFBP1 functions”
As we mentioned earlier, a previous study (Ref 24) has reported that other lysine residues on UFBP1 can also act as UFM1 accepting sites in vitro. However, investigating the significance of UFBP1 ufmylation on these additional lysine residues in vivo would require the generation of additional knockin mouse models carrying individual and combinations of these point mutations. Therefore, it goes beyond the scope of our current study. Unfortunately, the mechanisms regulating protein modifications on alternative residues, regardless of ufmylation or ubiquitination, remain poorly understood.

Reviewer 2 Report
Ubiquitin and ubiquitin-like proteins are ligated to their targets by highly specific and dynamically regulated ligases and can be removed by regulated and target specific deubiquitinases or related enzymes. Ubiquitination and related processes regulate a myriad of cellular processes, many involved in disease. Unraveling the precise targets and their regulation is a monumental task for science with the – already partially realized - promise of substantial benefits for medicine.
In this paper, the authors have investigated the physiological effects of disrupting Ufmylation of UFBP1 at position K268 in mammals using a knock-in mouse model. They find that removal of the Ufmylation site K268 on UFBP1 in mice does reduce the overall Ufmylation of proteins in fibroblasts. This is not uncommon in the ubiquitination field, where that activity of ligases can be tuned by modification. They go on to show that against expectation, the cellular stress response is not affected by the K268R mutation. The KI mice are, again against expectation, fertile and healthy, showing no abnormalities in heart and digestive system. In summary, the authors prove that the K268 Ufmylation site controls Ufmylation, but is not essential to the cellular stress response, mammalian development and heart or digestive system function. This is a novel and valuable result and furthers the understanding of Ufmylation.
The paper is very well written, the hypothesis and results are lucidly presented. The figures are well-made, but panels should be consistently ordered. This work does not require further language editing if minor typographical errors are corrected (see below). I fully support publication of this paper, if the concerns listed below are addressed.
Major issues:
This is less of an issue, but rather a suggestion: The authors state that Ufmylation is reduced when UFBP1 Ufmylation on K268 is disrupted. UFBP1 is likely a subunit in a larger complex (PMID: 36121123), which includes the ligase UFL1; the AlphaFold model places K268 at the interface between the two subunits, suggesting that modification could have functional impact. The authors could discuss their very important result in the context of the structural model of the UFBP1-UFL1 complex if they agree that this would add to the interpretation. While the K268R mutation is conservative, it would be beneficial to carefully inspect the model to ensure that the mutated residue is not directly involved in assembly of the complex.
According to the text (line 130), two ER stress inducers were used, one of them tunicamycin. The other one is not mentioned and only shows up as TG in the figures where data for it is presented, but not annotated in the figure legends. The materials and methods section does not provide any evidence about the identity of this mysterious compound either. The authors either need to add the missing information or remove the data shown for the second compound.
The authors describe the generation of the knock-in mouse strain in detail in the main text of the paper, which I appreciate very much. They introduce two mutations on the DNA level: one leading to the desired mutation K268R and another silent mutation (called nonsense mutation by the authors) in the triplet coding for D265. The authors present the introduction of the silent D265 mutation as intentional. What is the purpose of the D265 mutation? Is there a technical reason for introducing it or did the authors find it challenging to introduce just the K268K mutation (as can be the case in CRISPR-Cas9) and went with the silent mutation as this was still closest to the desired goal? I also do not understand why the authors call D265 a nonsense mutation, as the sequence is preserved on the protein level. As the authors are without doubt aware, in genetics, nonsense mutations introduce a premature stop codon, leading to the expression of truncated, likely unfunctional protein. A mutation that does not result in a change on the protein level would be called a silent mutation. Silent mutations can have substantial influence by e.g. disrupting splicing, so if they are introduced, correct expression of the protein has to be ascertained, as the authors have done with results presented in Fig. 1C. So, in summary, I would like the authors to explain the rationale for introducing the D265 silent mutation and apply correct molecular genetics terminology to render this part of the experimental description less confusing.
Minor issues:
Author list: why are the academic titles included? I think that is rather unusual and not necessary.
Figure 3D: Hoechst
186: annals might not be the right word
Author Response
Reviewer 2
Ubiquitin and ubiquitin-like proteins are ligated to their targets by highly specific and dynamically regulated ligases and can be removed by regulated and target specific deubiquitinases or related enzymes. Ubiquitination and related processes regulate a myriad of cellular processes, many involved in disease. Unraveling the precise targets and their regulation is a monumental task for science with the – already partially realized - promise of substantial benefits for medicine.
In this paper, the authors have investigated the physiological effects of disrupting Ufmylation of UFBP1 at position K268 in mammals using a knock-in mouse model. They find that removal of the Ufmylation site K268 on UFBP1 in mice does reduce the overall Ufmylation of proteins in fibroblasts. This is not uncommon in the ubiquitination field, where that activity of ligases can be tuned by modification. They go on to show that against expectation, the cellular stress response is not affected by the K268R mutation. The KI mice are, again against expectation, fertile and healthy, showing no abnormalities in heart and digestive system. In summary, the authors prove that the K268 Ufmylation site controls Ufmylation, but is not essential to the cellular stress response, mammalian development and heart or digestive system function. This is a novel and valuable result and furthers the understanding of Ufmylation.
The paper is very well written, the hypothesis and results are lucidly presented. The figures are well-made, but panels should be consistently ordered. This work does not require further language editing if minor typographical errors are corrected (see below). I fully support publication of this paper, if the concerns listed below are addressed.
We appreciate this Reviewer for the encouraging comments.
Major issues:
This is less of an issue, but rather a suggestion: The authors state that Ufmylation is reduced when UFBP1 Ufmylation on K268 is disrupted. UFBP1 is likely a subunit in a larger complex (PMID: 36121123), which includes the ligase UFL1; the AlphaFold model places K268 at the interface between the two subunits, suggesting that modification could have functional impact. The authors could discuss their very important result in the context of the structural model of the UFBP1-UFL1 complex if they agree that this would add to the interpretation. While the K268R mutation is conservative, it would be beneficial to carefully inspect the model to ensure that the mutated residue is not directly involved in assembly of the complex.
This is an excellent idea that we did not consider previously. If K268 does indeed reside at the interface between UFBP1 and UFL1 as predicted by AlphaGold, ufmylation at K268 could potentially facilitate their interaction. To support this concept, it would be beneficial to present the structural model predicted by AlphaGold and describe how ufmylation at K268 could potentially impact their interaction. However, due to our limited expertise in structural biology, we are unable to accurately explain the model. Therefore, we have decided to omit it from this manuscript. We hope the reviewer understands and agrees with this decision.
According to the text (line 130), two ER stress inducers were used, one of them tunicamycin. The other one is not mentioned and only shows up as TG in the figures where data for it is presented, but not annotated in the figure legends. The materials and methods section does not provide any evidence about the identity of this mysterious compound either. The authors either need to add the missing information or remove the data shown for the second compound.
We apologize for the confusion. Only TM was used. This is now corrected.
The authors describe the generation of the knock-in mouse strain in detail in the main text of the paper, which I appreciate very much. They introduce two mutations on the DNA level: one leading to the desired mutation K268R and another silent mutation (called nonsense mutation by the authors) in the triplet coding for D265. The authors present the introduction of the silent D265 mutation as intentional. What is the purpose of the D265 mutation? Is there a technical reason for introducing it or did the authors find it challenging to introduce just the K268K mutation (as can be the case in CRISPR-Cas9) and went with the silent mutation as this was still closest to the desired goal? I also do not understand why the authors call D265 a nonsense mutation, as the sequence is preserved on the protein level. As the authors are without doubt aware, in genetics, nonsense mutations introduce a premature stop codon, leading to the expression of truncated, likely unfunctional protein. A mutation that does not result in a change on the protein level would be called a silent mutation. Silent mutations can have substantial influence by e.g. disrupting splicing, so if they are introduced, correct expression of the protein has to be ascertained, as the authors have done with results presented in Fig. 1C. So, in summary, I would like the authors to explain the rationale for introducing the D265 silent mutation and apply correct molecular genetics terminology to render this part of the experimental description less confusing.
We are sorry for the confusion. Indeed, there is a technical reason to introduce this synonymous mutation encoding D265. This strategy can avoid additional mutations caused by inaccurate cleavage of the edited DNA by Cas9 and is quietly commonly used in the field. We have now added the description in Lines 94-97 as follows.
Specifically, we designed a specific guided RNA that introduces a missense mutation (A àG, mutating K268 to R) to the Ufbp1 locus and a synonymous mutation (CàT, encoding D265). The latter can avoid inaccurate re-cleavage of the edited DNA by Cas9.
Minor issues:
Author list: why are the academic titles included? I think that is rather unusual and not necessary.
We appreciate the reviewer for pointing out the mistake. The authors’ titles are now removed.
Figure 3D: Hoechst- done
“Hoechst 33, 342” is now corrected to “Hoechst 33342”.
186: annals might not be the right word- done
“Annals” is now corrected as “anus”.
Reviewer 3 Report
Tandra et al, engineered to delete a UFMylation site on UFBP1 in mice and tested several phenotypes. Despite the results being "negative", I believe that it is essential for the scientific community to know about them.
I only have a minor comment:
The last two paragraphs of the abstract sound very similar. Maybe one was supposed to be deleted during editing?
English quality is fine.
Author Response
Reviewer 3
Tandra et al, engineered to delete a UFMylation site on UFBP1 in mice and tested several phenotypes. Despite the results being "negative", I believe that it is essential for the scientific community to know about them.
I only have a minor comment:
The last two paragraphs of the abstract sound very similar. Maybe one was supposed to be deleted during editing?
We are sorry for the mistake. The redundant sentences (original Lines 38-41) are now removed.
We also removed some words in line 100-101
validated by genotyping and further